

# An attention embedded DUAL-LSTM method for financial risk early warning of the three new board-listed companies

Xiaojing Cheng

School of Accounting, Wuxi Taihu University, Wuxi, China

## ABSTRACT

Computer and financial fields are both involved in the interdisciplinary topic of financial risk early warning. We suggest an attention-embedded dual Long Short Term Memory (DUAL-LSTM) for the financial risk early warning to deal with the potential and constraints of rapid economic development to improve the precision of the financial risk prediction for the listed businesses on the New Third Board. First, feature fusion attentionally quantifies data characteristics, increasing the robustness and generalizability of data features. The model's predictive power is then increased by creating a dual LSTM model to meet the financial risk. The studies show that the attention-embedded dual LSTM model can achieve 96.9% of the F value scores and is superior to state-of-the-art model (SOTA) such as the Z-score model, Fisher discriminant method, logistic regression, and Back-Propagation network, achieves the advantage of time series in financial risk prediction. Additionally, for predicting financial risk, our algorithm performs flawlessly and effectively.

# INTRODUCTION

New three board-listed companies' financial risk prediction is a hot research point in the economic field. The financial risk early warning is to establish an early warning mechanism to predict the crisis for high-risk institutions with the intermittent change of economic and financial.

The prediction models of the early financial risk are often math models that calculate by traditional statistics. But these models always consist of linearization modules with strict limitations, which draws the nonlinear relationship within the financial risk difficulty and always obtains the lousy performance to be far away from the actual financial warning research. Later, methods, such as support vector machines (SVM), neural network (NN) and random forest (RF), have been exploited widely to forecast economics and finances by the rapid development of pattern recognition and deep learning, which can deal with complex nonlinear issues quickly and effectively by the nonlinear enhancing modules (*Kingma & Ba, 2014*; *Ahmad et al., 2021*). Among them, SVM is the most popular method, which can easily achieve the optimal strategy in hyperplane through the minimization of risk with perfect generalization. *Huang et al. (2015)* use SVM to construct KPCA-SVM

Corresponding author
Xiaojing Cheng, chengxj@wxu.edu.cn

financial crisis, early warning model. *Chun & Yang (2018)* find that SVM with polynomial kernel function has excellent prediction performance and learning ability to predict financial risks. As we know, predicting financial troubles is a classification task with a time sequence. However, SVM and NN are both traditional methods with invalid features and can't learn sequence features effectively, such as time or steps.

In recent years, deep learning has occupied various fields in the community of scholars through the continuous development of computer technology. The advantage of deep learning lies in the gradual learning of multiple networks to extract complex and compelling features (*Ni et al., 2014*), which can help models increase the prediction's accuracy. *Dixon, Klabjan & Bang (2015)* use DNN to predict the futures price of the Chicago Board of Trade and demonstrate that DNN can simulate the financial time sequence to obtain a better prediction result. *OuYang, Huang & Yan (2020)* find that LSTM can effectively capture the difference between the features with different economic time sequences of the Dow Jones Industrial Index. Deep learning is exploited narrowly for financial risk prediction because selecting parameters is hard, which effectively influences the prediction model. According to the optimizing strategies, such as cross-validation and reinforcement learning, *Marso & Merouani (2020)* significantly improve the prediction performance of financial risks and achieve the 90.30% of F value by optimizing the feedforward network with the CSA algorithm. The studies above directly input the data and do not pay attention to the impact of abnormal data on financial risks. In addition, the financial risk of enterprises listed on the New Third Board is affected by time, location, *etc.* Therefore, we must not only consider the prediction accuracy but pay attention to the time sequence of the model.

To solve the mentioned problems and to enrich cycle application areas of the neural network algorithm, we propose an attention-embedded dual LSTM model to predict the early financial risk for the three board-listed companies, which can optimize the network model to adapt the task of financial warning, improve the robustness of the model and increase the accuracy of the New Third Board listed companies financial risk prediction. It is conducive to the empirical analysis of the actual economic situation for the companies listed on the New Third Board.

Main contributions of our article include:

To address the issue of the time sequence for financial risk prediction, we propose an attention embedded dual LSTM model to determine the relationships among the attention features from the financial situation of the New Third Board listed companies.

We propose a dual LSTM framework to classify the financial situation to risk or safety. The first layer of LSTM is to attend the features and the second layer is to obtain the classification by inputting attended financial features.

## RELATED WORK

The task of financial risk early warning is to identify whether the financial risks will come in the future. We often propose a discriminant model to help people to obtain an enterprise's future financial situation with the previous financial or non-financial elements. Since then, the financial early warning model has included parts: (a) the appropriately selected analysis

methods; (b) analysis for determining samples; (c) the design of the application system, which uses some methods to analyze the future financial situation and get the situation results (*Fang Qi, 2021*).

Discriminant analysis (*Huang, 2021*; *Ahmad et al., 2022*), also known as linear discriminant analysis, is a typical representative of statistical analysis. Discriminant analysis is to establish specific discriminant criteria (discriminant functions) with the general condition of the classification results of the predicted objects and the values of multiple characteristic variables that can affect the classification results of the objects so that it can reduce the error probability of classifying the newly observed objects that have not appeared in the past by the discriminant criteria.

The process of the discriminant analysis can be shown as formulas:

$$J(w) = \frac{|\tilde{\mu}_1 - \tilde{\mu}_2|}{\tilde{s}_1^2 - \tilde{s}_1^2} \tag{1}$$

$$\tilde{s}_i^2 = \sum_{y \in w_i} (y - \tilde{\mu}_i)^2. \tag{2}$$

Logistic regression is also a multivariable model. The main difference between Logistic regression and linear regression is that the explanatory variable Y of linear regression is numerical, while that of Logistic regression is bivariate. The Logistic regression model classifies the samples according to the value of the input variables and divides the samples into Logistic binomial regression and multinomial Logistic regression according to the number of classified categories. The difference lies in whether the input samples are divided into two classes or multiple categories.

The process of the logistic regression can be shown as formulas:

$$\prod_{i=1}^{i=k} h(x_i) = \prod_{i=k+1}^{n} (1 - h(x_i)) \tag{3}$$

$$\prod_{i=1}^{i=k} h(x_i)^{y_i} (1 - h(x_i))^{1-y_i} \tag{4}$$

$$L(w) = \frac{1}{n} \sum_{i=1}^{n} -y \ln(h(x_i)) - (1 - y_i) \ln(1 - h(x_i)). \tag{5}$$

Artificial neural networks (ANNs) are used in this article. ANNs can simulate the human brain and the thinking system to process various information and features. There are many famous models, such as the back propagation (BP) network, which are networks that process information for intelligent tasks. The most popular is the BP network which consists of the feedforward network and loss function by error backpropagation. Its key point is to use the gradient search method to find the minimization of the difference between the ground truth and predicted output. It can process the tasks such as classification, detection, recognition and so on.

The prediction of early financial risk is a classification problem whose results are generally divided into two categories: with financial risk and without financial risk. It is a classification problem and is the most suitable scenario for applying neural networks. Therefore, it is feasible to predict early financial risk using neural networks for the listed companies on the New Third Board.

The financial data of an enterprise is continuous in time. For general neural networks, the data sets processed do not have the characteristics of time series, but the financial data of enterprises are constant in time. The report items affect each other, and the financial information of the previous year will also impact the following year. In particular, the ratio information, such as the year-by-year growth rate and the growth rate compared with the beginning of the period, are also critical financial data. Therefore, it is necessary to add time series based on a neural network and the recurrent neural network (*Zaremba, Sutskever & Vinyals, 2014*) to process the sequence information better.

## DUAL-LSTM EMBEDDED ATTENTION MODEL

This article proposes a financial risk early warning method for enterprises listed on the New Third Board based on the double-layer LSTM model. The technique takes LSTM as the primary processing unit. It mainly includes three parts: financial data processing, financial data saliency enhancement and financial risk judgment of enterprises listed on the New Third Board (*Hochreiter & Schmidhuber, 1997*).

### LSTM architecture

We apply the gradient decrease strategy in connection weights training to optimize the LSTM network. Considering containing the rapidly disappearing input information in the reverse transmission cycle sequence and the loss of information, we currently apply the LSTM network (*Yan & Ma, 2011*), which is the most widely used network architecture. LSTM can store all the existing information by building a memory function according to the structure of a recurrent neural network (RNN). The memory function is an essential principle for the LSTM model to solve the gradient vanishing problem.

Based on RNN, LSTM introduces a storage unit and gated storage unit to store historical information, which can realize the long-term storage state of helpful information. There are three types of gating in the basic unit of LSTM, including input gate, forget gate and output gate. By working together, the information can be stored in the memory unit memory at a step. The structure of neuronal cells in LSTM is presented in Fig. 1.

LSTM learning comes from the RNN, the new learning process is the need to update the weight and memory units, namely $c_t$, when the time step t. Set the LSTM hidden layer of input and output vector to $x_t$ and $h_t$, respectively. To control the current input data $x_t$ stored in the memory cell $c_t$ of the amount of information, an input gate is implemented, which is how much information can be saved to $c_t$. The formula is shown as follows:

$$i_t = \sigma \left( W_{xi} x_t + W_{hi} h_{t-1} + b_i \right). \tag{6}$$

The forgetting gate is a critical component of LSTM. Its function is to control what information is retained and what is forgotten and to give up unwanted information. It can

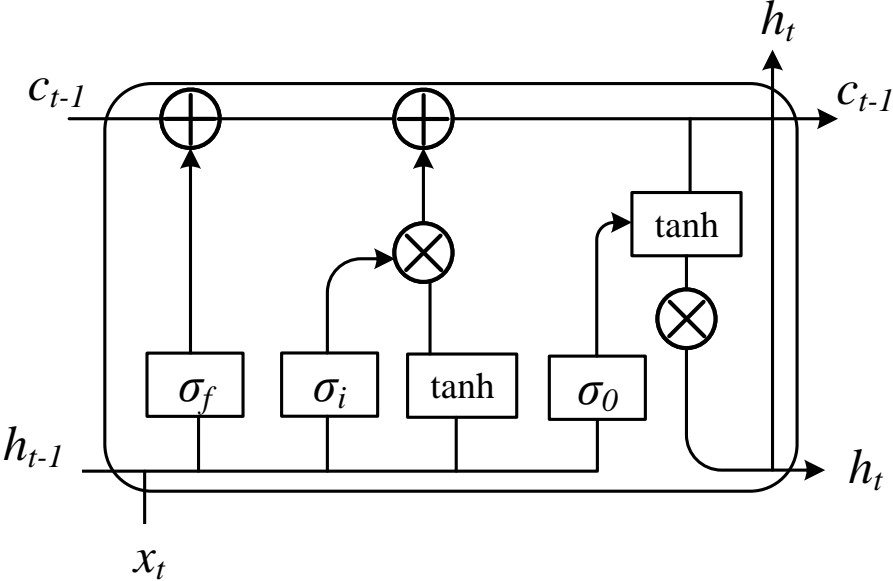

**Figure 1  The framework of the LSTM basic cell.**

mainly play the role of dimensionality reduction and avoid the gradient backpropagation caused by the gradient disappearance. At the same time, time change and the failure of convergence lead to the gradient explosion problem, which can then converge loss function better. The forgetting gate controls the self-connected unit and determines the discarded historical information to achieve the purpose of dimension reduction. Its core is to select the information in the memory unit at the last moment, which has an impact on the current memory unit:

$$f_t = \sigma \left( W_{xf} x_t + W_{hf} h_{t-1} + b_f \right) \tag{7}$$

$$c_t = f_t \cdot c_{t-1} + i_t \cdot \tanh \left( W_{xc} x_t + W_{hc} h_{t-1} + b_c \right). \tag{8}$$

We control the memory unit $c_t$ on the current output value $h_t$ to construct the output gate, which part of the memory unit will output in time step t:

$$o_t = \sigma \left( W_{xo} x_t + W_{ho} h_{t-1} + b_o \right) \tag{9}$$

$$h_t = o_t \cdot \tanh \left( c_t \right) \tag{10}$$

where W is the linear matrix and b is bias.

## Dual LSTM embedded attention model

Aiming at the related problems of financial risk prediction for enterprises listed on the New Third Board, this article proposes an attention-embedded dual LSTM model. The model is divided into two modules: Attention mechanism and double-layer LSTM architecture, as shown in the figure.

### Attention

There are two methods to classify financial state based on LSTM hidden state: One is to take the final hidden state of LSTM as the input of the classifier, which will lose some information far from the final hidden state; the other is to sum the hidden states of all time steps and then take the average value as the input of the classifier, but it cannot distinguish the influence of the input information of each time step on the composition classification. Therefore, we decide to use the attention mechanism to learn the importance of the input information at each time step to capture the critical information in many financial states and increase the weight.

Regard the input vector as LSTM $x_0, x_1, \ldots, x_t$, and the hidden state corresponding to the input, that is, vectors are $h_0, h_1, \ldots, h_t$ respectively. We then calculate the similarity $S_j$ between the current hidden state $h_j$ and the final hidden state $\bar{h}$ by using an attention mechanism. The formula is as follows:

$$S_j = \tanh\left(W_n h_j + b_n\right) \tag{11}$$

where $W_n$ and $b_n$ are the linearization matrix and bias. At each time step, the weight $a_j$, which is to process input information, is shown in the following formula:

$$a_j = \frac{\exp\left(S_j \times \bar{h}\right)}{\sum_{m=0}^{t}\left(S_j \times \bar{h}\right)}. \tag{12}$$

The hidden state vector $v$ of LSTM after adding attention weight is shown in the following formula:

$$v = \sum_{m=0}^{t}\left(a_j h_j\right). \tag{13}$$

### Dual LSTM embedded attention

Aiming at the financial feature representation $x_0, x_1, \ldots, x_t$ of enterprises listed on the New Third Board, this article proposes a novel dual LSTM to generate attended vector and warning classification results, respectively. The dual LSTM can be divided into the attention LSTM and the classification LSTM, respectively, whose basic framework is shown in Fig. 2.

In each time step, the attention LSTM includes the output of the particular classification LSTM at the previous time, the financial feature vector and the generated financial status encoding feature, respectively. The specific representation is shown in the following formula:

$$h_t^1 = \text{LSTM}_{\text{att}}\left(h_{t-1}^2, x_t\right) \tag{14}$$

where $h_{t-1}^2$ is the output of the Classification-LSTM at the previous time step. These inputs provide the status information of the classified LSTM, the context information of the financial state and the maximum context information of the generated financial state, respectively.

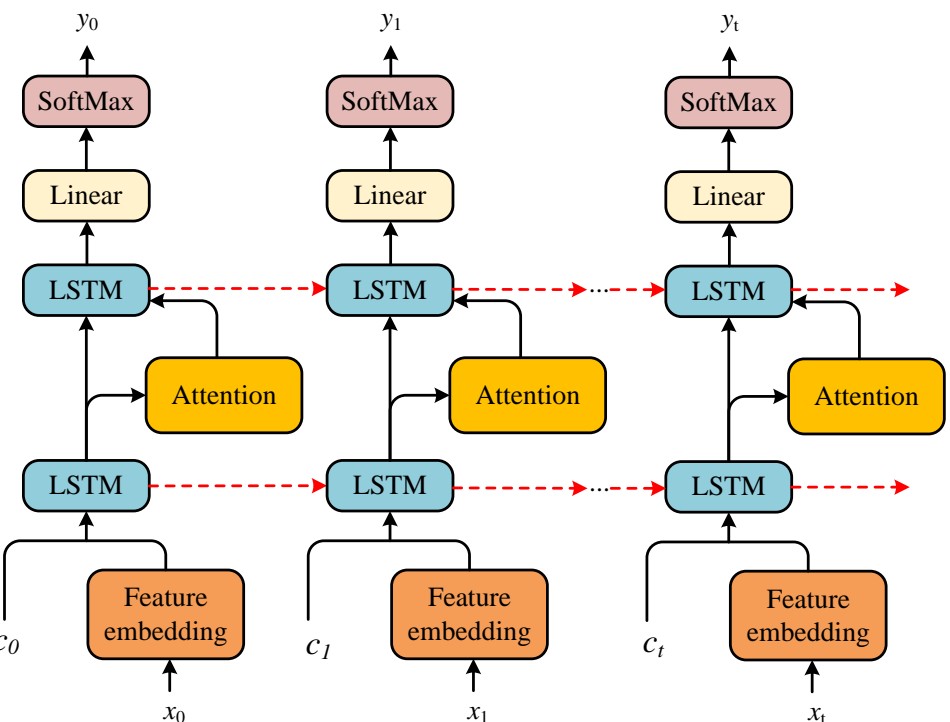

**Figure 2** The framework of the LSTM basic cell.

After obtaining the output $h_t^1$ of the attention LSTM at time t. At each time step, the attention weight $a_{j,t}$ is generated for each financial feature $x_t$:

$$a_{j,t} = W^T \tanh\left(W_x x_t + W_h h_t^1\right) \tag{15}$$

where $W^T, W_x, W_h$ are learnable matrixes. Then, according to Eq. (8), the financial features $\tilde{x}_t$ are input to the classification LSTM. After attention financial features of LSTM are obtained, the classification results y are generated by combining the output of the classification LSTM.

$$h_t^2 = LSTM_{cls}\left(\tilde{x}_t, h_t^1, h_{t-1}^2\right) \tag{16}$$
$$y = softmax\left(W_{cls} h_t^2 + b_{cls}\right) \tag{17}$$

where $W_{cls}$ and $b_{cls}$ are learnable linear transformation matrix and bias, respectively.

The financial indicators adopted by the model training are shown in Table 1.

## EXPERIMENT AND ANALYSIS

### Dataset

Considering the availability of data, our article selects listed companies on the New Third Board to provide a research idea of financial data of a specific enterprise type. The 2014–2018 public financial data of 100 enterprises are selected as the explanatory variables

**Table 1  Financial indicators.**

| | |
|---|---|
| Profitability | Total assets compensation |
| | Net asset income |
| | Net profit from sales |
| | Gross sales profit |
| Solvency | Asset liability ratio |
| | Equity ratio |
| | Tangible assets/net debt |
| Cash flow | Cash received from selling goods and providing services |
| | Cash from operating activities |
| | Net profit (cash) |

**Table 2  Training setting.**

| parameter | The values |
|---|---|
| Learning rate | 0.002 |
| Dropout | 0.3 |
| Model optimizer | Adam |
| Maximum sequence length | 6 |
| Number of Epoch | 200 |
| Batch size | 50 |

of the model for the experiment. The 2018 financial data is taken as the actual value and the 2014–2017 data is regarded as the training sample.

## Implement the details

We implement our experiments with the i7-12900k Cpu, 3080ti Gpu and the Pytorch deep learning framework.The parameter settings of the attention embedded dual LSTM model are shown in Table 2.

In this article, we evaluate the performance of the early financial risk prediction by standard classification accuracy rate. Considering the unbalanced samples, we apply the evaluation index to represent the prediction performance. The classification accuracy is Recall (R), Precision (P) and F. The sample measure standard F is quoted to evaluate the classification of unbalanced samples. If F is large, it indicates the performance of the financial risk prediction is superior and vice versa. The formulas of the evaluation metrics are presented by:

$$R = \frac{TP}{TP + FN} \tag{18}$$

$$P = \frac{TP + TN}{TP + FP + TN + FN} \tag{19}$$

$$F = \frac{2 \times \frac{TP}{TP+FP} \times \frac{TN}{TN+FN}}{\frac{TN}{TN+FN} + \frac{TP}{TP+FP}} \tag{20}$$

**Table 3  Comparative experiments with methods based on the deep learning framework.**

| Method | R | P | F |
|---|---|---|---|
| Z-score | 77.94% | 76.47% | 77.20% |
| Fisher discriminant | 82.35% | 89.71% | 85.87% |
| Logistic | 92.65% | 94.12% | 93.38% |
| BP neural network | 91.30% | 93.18% | 92.23% |
| Ours | 93.33% | 95.36% | 94.51% |

## Results and discussion

### Comparison results of different models

To evaluate our proposed model in this article, several classical financial early warning models are selected for experimental comparisons, such as the Z-score model (*Yan & Ma, 2011*), Fisher discriminant method (*Huang, 2021*), logistic regression (*Liguo et al., 2013*) and BP network on the same dataset. We present the results as shown in Table 3.

Because this article focuses on the early warning of financial risk, the special treatment (ST) companies' identification into the recalling rate of ST companies is even more critical. As seen above, the dual LSTM embedded Attention model has better performance, and the dual LSTM model can achieve a value of more than 90% in terms of recall, precision and F measure, which is a specific identification model of ST and the l performance of the classifier is better. First, the F value of the attention-embedded dual LSTM is increased by more than 18% compared with the original Z-score because all the model structures designed in this article are ahead. Secondly, our method improves the F value by more than 5% to the Fisher's discriminant in all three evaluation indexes. Then, the effect of these two methods, Logistic and BP neural network, are significantly improved, which are still inferior to our proposed method. Finally, the proposed method achieves the best experimental results by embedding the Attention module and creatively using the two-layer LSTM as a financial warning.

### Influence of hidden layer size and layers number

To discuss the number of LSTM and the size of hidden layers, we set the batch size to 4 and epochs =50 during training to experiment. In the experimental process, we will complete the determination of hyperparameters by comparing the recognition accuracy during training.

The dimension size of the hidden layer can be expressed as a bar graph for the model training results (Acc refers to the correct rate of recognizing ST into ST). As shown in Fig. 3, we can find that the model's performance continues to rise with the increase of dimension until the hidden layer dimension reaches a peak of 1024, where the model achieves the best training accuracy. Therefore, we choose the dimension of the hidden layer to be 1024.

The number of layers can be represented as a histogram (Acc refers to the correct rate of recognizing ST into ST). As shown in Fig. 4, we can also find that the model's performance continues to rise with the number of model layers increasing. When the length of the LSTM

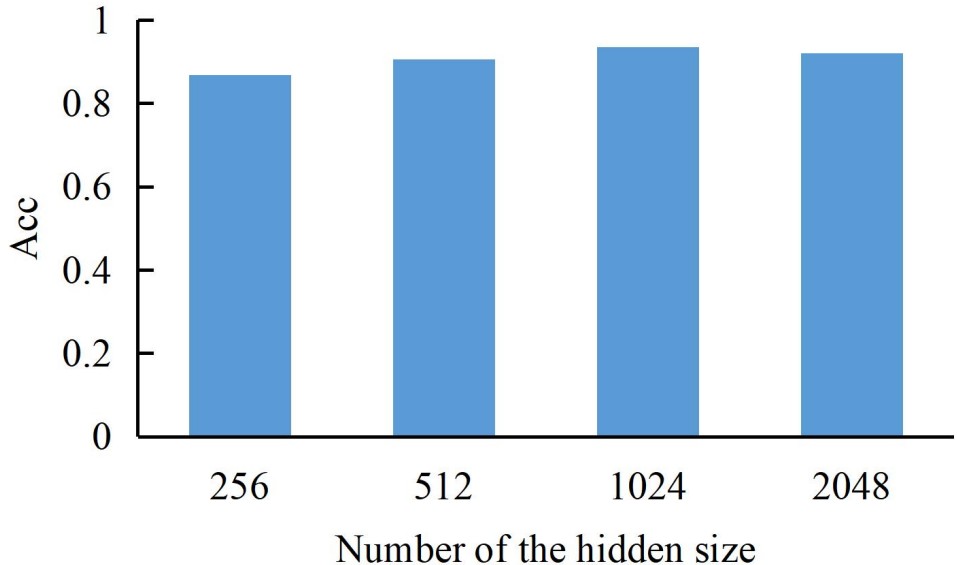

**Figure 3** The accuracy of our method with a different dimension of LSTM hidden layer.

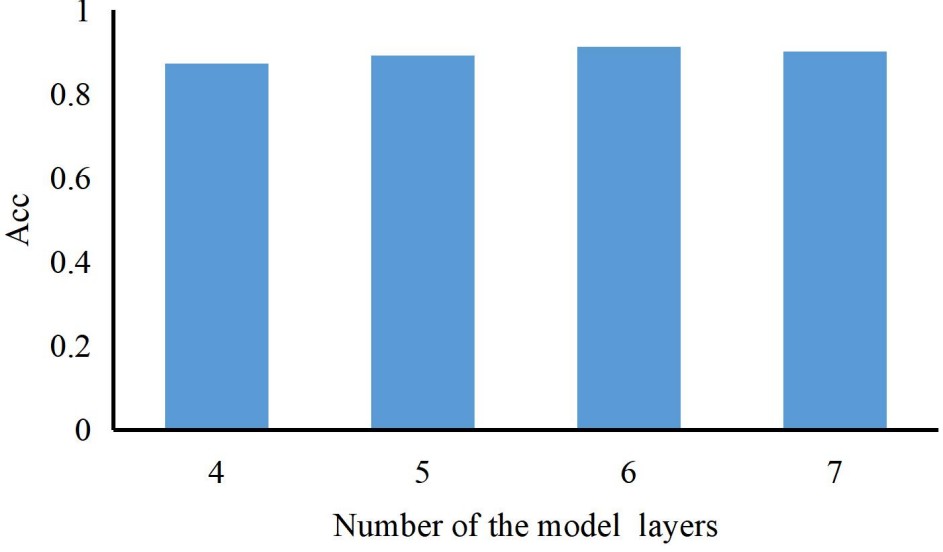

**Figure 4** The accuracy of our method with a different number of LSTM hidden layers.

is 6, our method achieves the best results and training accuracy. So, we set the number of the model layer to 6.

The above results directly affect the training process's rigor and positively affect the financial risk prediction of the Three New Board listed enterprises.

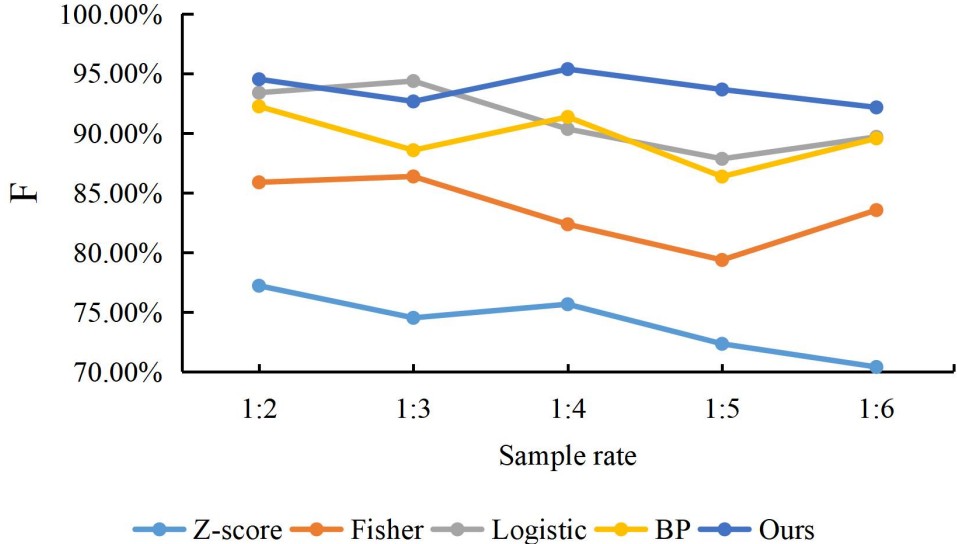

**Figure 5** The accuracy of methods with different sample rate.

### Comparison of prediction stability

To compare the predictive stability of our model with the Z-score model, Fisher discriminant method, logistic regression and BP neural network model Since the 1:1 of the rate can affect the randomness. This article changes the proportion between positive and negative samples by adding more positive samples corresponding to the negative samples one by one to investigate the robustness of the dual LSTM embedded Attention model. Specifically, the ratio between the financial risk sample and the safe financial sample is reset, following the ratio of 1:2, 1:3, 1:4, 1:5 and 1:6, respectively. Then, we apply these five proportions to structure the five types of models, which are to evaluate the prediction stability for the financial risks. Because the samples, which are chosen with the five proportions, are not balanced samples, we exploit a special measurement F for the non-balanced samples to obtain the performance of the prediction. The experiment results are presented in Fig. 5.

We can find that the F value of our proposed model is higher than other models with any proportions of positive and negative samples, which indicates that Z-score, Fisher discriminant, Logistic regression and BP network are inferior to our method for both financial risk samples and safe samples. Graphically, the F value of our proposed model is relatively more stable in trend and less volatile, which indicates that the prediction of the financial risk samples and safe samples are accurate with different proportions.

## CONCLUSION

Financial risk early warning has been one of the hot spots research. Considering the complexity of the financial risk prediction of the New Three Board listed companies, this article proposes an attention embedded dual LSTM model to predict financial early

risk, which can be successfully applied to New Three Board listed enterprise financial risk forecast. Firstly, the financial features of listed enterprises on the New Three Board are fused and we quantify them to input the attention embedded dual LSTM model. Then, a new embeddable Attention module is designed to enhance the representation ability of financial features. Finally, a dual localization algorithm based on a spring model (LASM) with embedded attention is designed to predict the financial risks of enterprises listed on the New Third Board. Compared with the other four models, the results can demonstrate that the dual LSTM model embedded Attention has better performance of prediction in terms of all evaluate metrics and the $F$ value is increased by 17.31%, 8.64%, 1.13% and 2.28%, respectively, where the accuracy is statistically significant. In the future, we will explore to extend the financial risk prediction to the all companies. Then, we will try to research on the attention to find out more important point into financial situation of companies and to attended them to increase the accuracy of prediction.

## ACKNOWLEDGEMENTS

We would like to thank the anonymous reviewers whose comments and suggestions helped improve this manuscript.

### Funding

This work was funded by Jiangsu Provincial Department of Science and Technology, project number BR2021007 and supported by the National Social Science Foundation Project, project number 19BJL020. The funders had no role in study design, data collection and analysis, decision to publish, or preparation of the manuscript.

### Grant Disclosures

The following grant information was disclosed by the author:
The Construction Plan of Scientific Research and Innovation Platform of Wuhan College: KYP202001.
Research and Innovation Team of Wuhan College: KYT201903.

### Competing Interests

The authors declare there are no competing interests.

### Author Contributions

- Xiaojing Cheng conceived and designed the experiments, performed the experiments, analyzed the data, performed the computation work, prepared figures and/or tables, authored or reviewed drafts of the article, and approved the final draft.

### Data Availability

   The code is available in the Supplemental File. The data is available at GitHub: https://github.com/hackingthemarkets/datasets.git.

## Supplemental Information

Supplemental information for this article can be found online at http://dx.doi.org/10.7717/peerj-cs.1271#supplemental-information.

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
