# Peer review of "An attention embedded DUAL-LSTM method for financial risk early warning of the three new board-listed companies"

_PeerJ Computer Science, doi:10.7717/peerj-cs.1271_

## Round 0.1 · original submission · Minor Revisions

Dear author,
Your paper has been reviewed by the experts in the field and you will see that it requires some improvements before we consider it. Therefore you are requested to update the article in line with what the reviewers has suggested and resubmit. Thank you

·

Basic reporting

no comment

Experimental design

no comment

Validity of the findings

no comment

Additional comments

New three board-listed companies' financial risk prediction has been one of the hot issues in economic field research. Currently, the research on the problem of low accuracy of financial risk recognition. The authors proposed an attention-embedded dual LSTM model to address this issue, which can optimize the network model to adapt to the task of financial warning, improve the robustness of the model and increase the accuracy of the New Third Board-listed companies' financial risk prediction.
It effectively solves the time sequence problem of financial features of enterprises listed on the New Third Board and helps to make responsible decisions in advance, which has good practical application value. After the following issues are resolved, I recommend accepting this article
1. What is the outstanding contribution of this paper? I don't know precisely what the actual problem solved in this article is from the introduction. In addition, the contribution should be introduced.

2. I suggest a definition of Financial Risk Early Warning at the beginning of the introduction so that readers can better understand the author's work.

3. In the related work section, I believe that the authors can list some core formulas to explain the references of discriminant analysis, logistic regression and so on.

4. Please explain why the authors don’t use Accuracy to evaluate the performance, which is more suitable for the classification task.

5. What are the specific framework technologies used in the design of the presentation layer, Pytorch or Tensroflow or Mxnet? It should be added to the Implement details section to help authors reproduce it easily.

6. The language of the conclusion needs to be further strengthened. The description of the author coincides with the abstract, and the author should focus on the outstanding contribution of the research and the application direction.

Reviewer 2 ·

Basic reporting

In this paper, An Attention DUAL LSTM Embedded Method for Financial Risk Early Warning of the Three New Board Listed Companies is proposed to improve the accuracy of prediction and understand the state of the companies’ financial. In practical application, the method in this paper can be used to reveal the financial risks of enterprises listed on the New Third Board, which is helpful to predict the financial status of enterprises and make response decisions in advance.
However, there are also some problems, some revisions needed to be revised to make sure that the manuscript can be accepted. The commonly problems are as follows:
1. The English of your manuscript should be revised in any section of the paper before resubmission. We strongly suggest that you obtain assistance from a colleague who is well-versed in English or whose native language is English.

2. I suggest that it is necessary to add the introduction of Financial Risk Early Warning in the first section, which can help readers to understand this paper. And segment the first paragraph into two parts.

3. As we known, there so many Financial Risk features. What’s the Financial Risk indicators is it? They are the sources of features and the input of the models.

4. The reason that the attention module is inserted between the dual LSTM should be introduced. In addition, the attention is an independent unit which can be applied to construct network, so shall we drop the first LSTM?

5. As I known, the Transformer and Bert can process the sequence task better. What reason is it to choose the LSTM?

6. The conclusion section should be revised carefully to concentrate on the contribution of this paper. Please check it and add the research plan for the Financial Risk Early Warning in the future.

Experimental design

In this paper, An Attention DUAL LSTM Embedded Method for Financial Risk Early Warning of the Three New Board Listed Companies is proposed to improve the accuracy of prediction and understand the state of the companies’ financial. In practical application, the method in this paper can be used to reveal the financial risks of enterprises listed on the New Third Board, which is helpful to predict the financial status of enterprises and make response decisions in advance.
However, there are also some problems, some revisions needed to be revised to make sure that the manuscript can be accepted. The commonly problems are as follows:
1. The English of your manuscript should be revised in any section of the paper before resubmission. We strongly suggest that you obtain assistance from a colleague who is well-versed in English or whose native language is English.

2. I suggest that it is necessary to add the introduction of Financial Risk Early Warning in the first section, which can help readers to understand this paper. And segment the first paragraph into two parts.

3. As we known, there so many Financial Risk features. What’s the Financial Risk indicators is it? They are the sources of features and the input of the models.

4. The reason that the attention module is inserted between the dual LSTM should be introduced. In addition, the attention is an independent unit which can be applied to construct network, so shall we drop the first LSTM?

5. As I known, the Transformer and Bert can process the sequence task better. What reason is it to choose the LSTM?

6. The conclusion section should be revised carefully to concentrate on the contribution of this paper. Please check it and add the research plan for the Financial Risk Early Warning in the future.

Validity of the findings

Overall, the findings stated and obtained are seems to be justified.

Additional comments

In this paper, An Attention DUAL LSTM Embedded Method for Financial Risk Early Warning of the Three New Board Listed Companies is proposed to improve the accuracy of prediction and understand the state of the companies’ financial. In practical application, the method in this paper can be used to reveal the financial risks of enterprises listed on the New Third Board, which is helpful to predict the financial status of enterprises and make response decisions in advance.
However, there are also some problems, some revisions needed to be revised to make sure that the manuscript can be accepted. The commonly problems are as follows:
1. The English of your manuscript should be revised in any section of the paper before resubmission. We strongly suggest that you obtain assistance from a colleague who is well-versed in English or whose native language is English.

2. I suggest that it is necessary to add the introduction of Financial Risk Early Warning in the first section, which can help readers to understand this paper. And segment the first paragraph into two parts.

3. As we known, there so many Financial Risk features. What’s the Financial Risk indicators is it? They are the sources of features and the input of the models.

4. The reason that the attention module is inserted between the dual LSTM should be introduced. In addition, the attention is an independent unit which can be applied to construct network, so shall we drop the first LSTM?

5. As I known, the Transformer and Bert can process the sequence task better. What reason is it to choose the LSTM?

6. The conclusion section should be revised carefully to concentrate on the contribution of this paper. Please check it and add the research plan for the Financial Risk Early Warning in the future.

---

## Round 0.2 · accepted · Accept

Good luck and thank you for your contribution